# Lead, Cadmium, and Arsenic in Raw Cow’s Milk in a Central Andean Area and Risks for the Peruvian Populations

**DOI:** 10.3390/toxics11100809

**Published:** 2023-09-25

**Authors:** Jorge Castro-Bedriñana, Doris Chirinos-Peinado, Elva Ríos-Ríos, Gianfranco Castro-Chirinos, Perfecto Chagua-Rodríguez, Gina De La Cruz-Calderón

**Affiliations:** 1Research Center in Food and Nutritional Security, Universidad Nacional del Centro del Perú, Huancayo 12001, Peru; dchirinos@uncp.edu.pe; 2Department of Chemistry, Science Faculty, Universidad Nacional Agraria La Molina, Lima 14024, Peru; erios@lamolina.edu.pe; 3Psychology Area, Teleperformance Perú, Lima 15076, Peru; gianfrancocc42@gmail.com; 4Faculty of Agroindustrial Engineering, Universidad Nacional Autónoma Altoandina de Tarma, Tarma 12701, Peru; pchagua@unaat.edu.pe; 5Quality Control Department, Universidad Nacional Autónoma de Chota, Cajamarca 06121, Peru; gdelacruz@unach.edu.pe

**Keywords:** health hazards, dietary risk, contamination level, permissible limits, risk assessment

## Abstract

Milk and its derivatives are basic foods in Peru, especially for children. The Junín region, in the central Andes, is one of the leading dairy basins. However, the safety of milk is affected by mining–metallurgical activities, wastewater dumping, organic residues, and inappropriate use of organophosphate fertilizers in agriculture whose contaminants reach the food chain, putting human health at risk. The purpose of this study was to evaluate the bioaccumulation of lead (Pb), cadmium (Cd), and arsenic (As) in milk produced on a representative farm in central Peru, which uses phosphorous agrochemicals and is adjacent to a small mineral concentrator and a municipal solid waste dump, and to evaluate the potential risk for the Peruvian population of 2–85 years considering three levels of daily intake by age, which constitutes the innovative contribution of the study. These three elements were quantified by flame atomic absorption spectrometry following standardized procedures. The mean contents of Pb (0.062 mg/kg), Cd (0.014 mg/kg), and As (0.030 mg/kg) in milk exceeded the maximum limits allowed by international standards. At all ages, the target quotient hazard followed a descending order of As > Pb > Cd, being > 1 in the case of As. The hazard index was >1 for children under 7, 9, and 11 years of age in the scenarios of low, medium, and high milk intake. The information is valid for formulating policies to prevent adverse health effects and develop standards and awareness programs, monitoring, and control of heavy metals in milk in Peru.

## 1. Introduction

Milk provides essential nutrients and energy for proper growth and development, and its consumption goes from infancy to old age. It is valued not only for its nutrient contribution but also that of other bioactive compounds [1]. In adults of North America, milk consumption amounts to approximately 267 L/year [2]. In Peru, consumption per person is 87 L/year, well above the European average of 59.4 L/year, while Germany reports 53.7 L/year [3].

It has been suggested that a high milk intake is associated with protective effects against coronary heart disease, stroke, diabetes, and hypertension. It also promotes good bone health, a strong immune system, and the prevention of dental caries, dehydration, respiratory problems, obesity, and osteoporosis, along with a decreased risks of colorectal and bladder cancer [1]. Depending on its innocuousness and safety throughout the production chain, there may be potential health risks from consuming milk with biological and chemical contaminants.

One of these problems is the presence of heavy metals and metalloids, which do not fulfill any essential function or provide any benefits for humans, and no homeostasis mechanism is known for them [4]. On the contrary, they are toxic even at relatively low levels; therefore, information is required on their concentrations in food, on the level of human exposure through food intake, and on the evaluation of potential health risks [4]. This information should be used by food safety agencies, which are responsible for ensuring that food for human consumption does not pose a health risk.

Lead (Pb), cadmium (Cd), and arsenic (As) are widely dispersed in the environment due to anthropogenic activities, including industrial emissions, traffic, the indiscriminate use of phosphorous agrochemicals, domestic waste dumped into rivers, other processes which ultimately reach the water and the soil through the air, causing changes in their natural composition, their dynamics, and their vegetation [5]. On entering the food chain, they affect human health [6,7,8], and due to their persistence in the environment, they can bioaccumulate in living organisms. Calcium ammonium nitrate fertilizer contains Pb up to 24.6 mg/kg, and NPK fertilizers have up to 12 mg/kg [9].

According to the United States Agency for Toxic Substances and Disease Registry, Pb, Cd, and As are among the most toxic elements [10], and it is well established that exposure to them can have a wide range of adverse health effects, including specific disorders [11,12]. Pb accumulates mainly in the bones and teeth, remaining for up to three decades, and its release is easy during pregnancy and lactation, affecting fetal development, and is detrimental to the neurocerebral development of young children. It is distributed throughout the body until it reaches the brain, liver, kidneys, and bones [13,14]. There is no level below which it can be said that lead exposure has no harmful effects. The Center for Disease Control and Prevention (CDC) uses a blood lead reference value of 3.5 micrograms per deciliter (µg/dL) to identify children with higher blood lead levels than is normal in most the children [15].

Cd is a potential threat to the environment because it biomagnifies in the food chain and is classified as a human carcinogen [16]. It is highly toxic to all living organisms and generally comes from phosphate fertilizers [17]. Phosphate-based triple superphosphate contains up to 53.2 mg/kg of Cd [18]. Cd toxicity causes adverse effects on most organ systems in humans and animals, leading to multisystem disorders and hematologic, neurologic, musculoskeletal, renal, hepatic, cardiovascular, pulmonary, and reproductive complications [16].

Arsenic is the most abundant metalloid in nature and is present in groundwater all over the world [19]. The principal source of anthropogenic exposure is industrial smelting, mainly of copper [20].

About 99% of As is absorbed through the intestinal mucosa and passed into erythrocytes. It interacts with the globin fraction of hemoglobin, accumulates in the liver, kidneys, heart, lungs, muscular and nervous system, gastrointestinal tract, and spleen. Its chronic intake can cause hyperkeratosis, hyperpigmentation, hepatotoxicity, nephrotoxicity, cardiovascular toxicity, reproductive toxicity, anemia, intestinal dysbiosis, and lung cancer [19,21]. As also contaminates various food sources, multiplying the net exposure to this metalloid [22].

The population in the central Andes of Peru has been exposed to emissions from mining–metallurgical activity for more than 100 years, whose vapors and dust travel in the air and are deposited in water and soil. Likewise, the high traffic on its roads and the misuse of fertilizers and agrochemicals based on phosphate rock in agriculture release heavy metals [23], a context that is also observed in many developing countries. Therefore, the products obtained from these soils must be monitored for their metal and metalloids content by evaluating the potential risk for their consumption, comparing them with toxicologically acceptable levels [4].

Located in the central Andes of Peru, the Mantaro basin is an important dairy area which has been exposed to mining–metallurgical contamination for more than ten decades, which generally settles in the headwaters of basins. Most agricultural production systems use irrigation water contaminated with heavy metals. In addition, the indiscriminate use of phosphorous agrochemicals in agrifood production increases contamination. The Pb, Cd, As, and other toxins are carried from the soil to pastures and livestock, and at the end of the food chain, humans consume these products.

The objective of this study was to determine the concentrations of Pb, Cd, and As in the soil, multi-species pastures, and raw milk from cows raised in an environmentally unsuitable area due to its proximity to a small mineral concentrator and to a municipal landfill, due to the use of irrigation water from the Mantaro River which is known to be contaminated by a cocktail of heavy metals and metalloids, and due to use of fertilizers based on phosphate rock. By estimating the weekly intake (WI), the objective risk coefficient (TQH), and the hazard index (HI) in the Peruvian population from 2 to 85 years old, considering three levels of daily milk intake (low, medium, and high for each age), we generate toxicological evidence of the milk quality in this region of the world. The fact that we present results for the entire Peruvian population in three scenarios of daily milk consumption by age argues for the originality of this study.

## 2. Materials and Methods

### 2.1. Study Period and Location

Soil, multi-species pasture, and milk sampling was carried out in April and May 2022 at the representative dairy farm located in the district of El Tambo in the province of Huancayo, Peru, at 3214 m above sea level, southern altitude 12°4′36.3″ S (−12.07675254000), and western longitude 75°13′30.2″ O (−75.22505777000), located on the right bank of the Mantaro River, whose pastures adjoin a small mineral concentrator that has been operating for more than 50 years and the former municipal dump “El Eden” (Figure 1) where inadequate final disposal of solid waste is carried out, the decomposition of which also affects the quality of air, water, and soils in the surrounding areas. The climate is characterized by two well-defined seasons, namely, the rainy season from October to March and the dry season from April to September. The region is characterized by a tundra climate, wherein the temperatures remain considerably low even during the warmest months. The climate is classified as ET according to Köppen and Geiger. In Huancayo, the mean yearly temperature amounts to 8.7 °C. In Huancayo; about 1682 mm of precipitation falls annually [24].

### 2.2. Animals and Breeding System

The Holstein cows used in the study had an average age of 4.7 years, and 12 clinically healthy cows were used. They had an average production of 9 L/day and were raised in a semi-extensive system with grazing during the day in *Dactylis glomerata* pastures, alfalfa (*Medicago sativa*), and with a portion of cut forage (*Hordeum vulgare*), all produced on the same farm. The irrigation of the pastures was carried out with water coming from a spring near the Mantaro River. This farm is representative of the study area, and the surrounding family farms receive advice from the technical staff of the selected farm.

### 2.3. Milk, Soil, and Multi-Species Pasture Sampling and Heavy Metal Analysis

After mechanical milking, to obtain 2 milk samples per cow, 24 samples of 250 mL of milk were collected from 12 cows (12 in April and 12 in May from the same cows), which were obtained in accordance with the milking protocol the Peruvian Technical Standard 202.112:1998 (revised in 2013) using first-use sterile polyethylene bottles previously washed with nitric acid and rinsed with bidi stilled water, which were labeled and kept at 4 °C for immediate shipment to the laboratory of the National Institute of Agrarian Innovation, Huancayo (INIA, Peru).

The sample’s milk preparation was carried out via the dry method and via acid mineralization; 50 g of each homogenized sample was placed in porcelain crucibles to be dried at 100 °C until reaching a constant weight. They were incinerated in a muffle furnace (Protherm 442-ECO110/15) at 450 °C/15 h, and after cooling them to 16 °C, they were treated with 2 mL of 2N HNO_3_. After evaporation and cooling, they were incinerated at 450 °C/1 h. The ashes were recovered with 20 mL of 0.1 N HNO_3_, filtered on Watman No. 40 paper, and stored in polypropylene tubes under refrigeration. High-purity reagents (Merck kGaA, Darmstadt, Germany) were used. For the quantification of Pb, Cd, and As, a flame atomic absorption spectrometer (NAMBEI AA320N) was used according to the AOAC 973.35 method, using wavelengths of 283.3 and 228.8 nm for Pb and Cd [27], respectively, and for As. of 193.7 nm [25]. The analyses respond to validated and standardized laboratory protocols; the precision and accuracy of the analytical methods follow ISO5725-2:1999. To elaborate the calibration curves, Sigma-Aldrich Pb 986 ± 4 mg/kg standards were used, and for verification of the precision of the analytical method, standard solutions of Pb, Cd, and As of 155 ± 0.04, 150 ± 0.05, and 50 were used, respectively, ± 0.02 mg/kg of milk [28]. The concentrations of their corresponding runs were 148, 152, and 49.6 mg/kg. The detection limits (LOD) of Pb, Cd, and As in milk were 0.03, 0.03, and 0.028 μg/L, respectively. The LOD of Pb, Cd, and As in forage were 2.40, 0.40, and 0.03 μg/kg, respectively. The LODs of Pb, Cd, and As for the soil were 0.1, 0.01, and 0.02 mg/kg, respectively.

To obtain information on the concentration of Pb, Cd, and As in the soil and multi-species pasture, in the intermediate period of milk collection, six soil samples and six grass samples were taken from the same sampling place. The 0.5 kg samples of the topsoil (0–20 cm depth) were collected using standardized procedures [29]; after one day of natural drying, they were crushed and sieved (2 mm mesh) to eliminate gravel, stones, and other impurities, and were subsequently homogenized, weighed, and hermetically sealed. The analyses followed the USEPA 3050B (SW-846) method. Quantification was also conducted via flame atomic absorption spectrometry, following the AOAC Official Method 975.03 protocol [30]. The corresponding grass samples were washed with tap water, removing dirt particles, rinsed with deionized water [31], dried at 70 °C, and finely ground. The digestion and quantification of the heavy metals in the pastures were similar to those of the soil. To evaluate the Pb and Cd according to the Environmental Quality Standards, 70 and 1.4 mg/kg were used, respectively [32]; for As, the standards of the Canadian Council Environment Minister were used, with a maximum of 12 mg/kg [33,34]. The maximum permissible limit (MPL) of Pb used for forage was 10 mg/kg of dry matter [35]; for Cd and As, values of 1 and 2 mg/kg, suggested by the Council of the European Parliament [36], were considered. Heavy metal concentrations in all study samples are expressed in mg/kg.

### 2.4. Estimated Weekly Intake (WI)

Exposure to Pb, Cd, and As was estimated using the mean concentrations in milk and the estimated intakes in the Peruvian population aged 2–85 years reported by the National Center for Food and Nutrition of the National Institute of Health [37]. The study included 62,600 individuals, being the only study that uses current population results to date. Three milk consumption scenarios were considered (low, medium, and high) for all ages, estimated based on information reported in different national and international studies, allowing continuous data generation for risk assessments [38,39,40,41]. The estimated weekly intake of each metal (WI: µg/week of milk consumption) was compared with the established Tolerable Weekly Intake (TWI) for each item [42,43,44].

### 2.5. Target Quotient Hazard (TQH)

The TQH for Pb, Cd, and As epidemiologically explains whether the heavy metals in food are within the level of the established limits and estimates the risk to human health [45,46,47,48]. To evaluate the TQH of chronic exposure to Pb, Cd, and As contained in milk, the following equation was used [49]:(1)$$\mathrm{TQH}=\frac{\left(EF*ED*Wmilk*Cmetal\right)}{RfD*Body\;Weight*TA},$$
wherein the following terms were applied:*Cmetal*: metal concentration in milk;*Wmilk*: daily milk intake;*EF*: exposure frequency (365 days per year;*ED*: the exposure period is equivalent to the mean longevity for an adult. For Peru, it is estimated in 76.5 years;*TA*: mean useful life time is 27,922.5 days;*RfD*: reference oral dose. For Cd, Pb, and As, values of 0.001, 0.0035, and 0.0003 mg/kg b.w./day in kg were used, respectively [50,51,52,53].

### 2.6. Hazard Index (HI)

The HI assesses the chronic hazard index with respect to human health for various toxic elements and represents the long-term cumulative risk. This was determined by adding the THQ of Pb, Cd, and As; if the HI is < 1, there is no risk to human health [7,8,54].

### 2.7. Data Analysis

To evaluate if the average contents of Pb, Cd, and As in the soil, grass, and milk exceed the MPL, *t*-tests were performed on a sample. The Pb, Cd, and As MPLs soil were 70, 1.4, and 12 mg/kg; for pastures, they were 30, 1.0, and 2.0 mg/kg; and for milk, they were 0.02, 0.0026, and 0.014 mg/kg, respectively. To evaluate the statistical differences between the concentrations of Pb, Cd, and As between the soil, grass, and milk samples, one-way ANOVAs were performed. Differences between means were evaluated using Tukey’s test, and *p* values < 0.05 were considered significant. The associations between each element in the soil-pasture and pasture-milk were estimated using Pearson correlations. WI, DCR, TQH, and HI calculations were made in µg/kg. The software used in the statistical processing was SPSS V26 (IBM, Endicott, NY, USA).

## 3. Results

### 3.1. Pb, Cd, and As Concentration in the Study Samples

Mean concentrations and standard deviation (SD) of Pb, Cd, and As in the soils were higher than in pastures and milk (*p* < 0.01) (Table 1).

Even when the average concentrations of Pb and Cd in the soil exceeded the LMP, the multi-species pasture had concentrations of Pb and Cd lower than the LMP. The As concentrations in soils and grasses did not exceed the MPL; however, the contents of the three elements in the milk exceeded the MPL (Figure 2), a dynamic that should be studied in greater depth.

Positive correlations were observed between the concentrations of the three elements. For Pb and Cd, r = 0.90, *p* < 0.01; for Pb and As, r = 0.78, *p* < 0.01; and for Cd and As, r = 0.82, *p* < 0.01; that is, at a higher level of Pb, a higher level of Cd and As was also found. These measurements establish a linear relationship between two variables and have made it possible to quantify the degree of joint variation between them; by increasing the concentration of Pb, the concentration of Cd and As increased, which would have a synergistic action in their bioaccumulation. The bioaccumulation percentage in descending order was As > Cd > Pb, with As being the major concerning element.

### 3.2. Weekly Intake (WI), Target Risk Quotient (TQH), and Risk Index (HI)

The WIs of Pb and Cd through the consumption of raw cow’s milk in the scenarios of minimum, medium, and maximum intake in the Peruvian population aged 2–85 years were below the TWIs established for these metals [40], while the WI for As was above the TWI established by USEPA [44] (Table 2). As the WI determines the TQHs of each toxic element, the TQHs for Pb and Cd at all ages and consumption levels were <1 (Figure 3 and Figure 4; Table 3). The TQH for As was >1 in children (Figure 5).

The TQH followed a descending order of As > Pb > Cd, with values in the range of 0.05–1.13, 0.01–0.28, and 0.01–0.24 for the minimum milk intake for the population aged 2–85 years; the values for a medium milk intake were between 0.08 and 1.41, 0.02 and 0.35, and 0.02 and 0.30, and the values for a high milk intake were between 0.13 and 1.69, 0.03 and 0.42, and 0.03 and 0.36, respectively. The HI values at minimum, medium, and maximum intake were between 0.08 and 1.65, 0.12 and 2.06, and 0.19 and 2.47, respectively, observing that they were >1 in children under 7, 9, and 11 years of age, and at each intake level of milk in adults, all HI values were <1.

To estimate the TQH and HI in children aged 2 to 5 years, minimum, average, and maximum daily milk consumption values of 0.40, 0.50, and 0.60 kg were considered; for people from 6 to 19 years, the values were 0.48, 0.60, and 0.72 kg; for people aged 20 to 59 years the values were 0.10, 0.15, and 0.23 kg; and for people aged 60 to 85 years, the values were 0.12, 0.19, and 0.28 kg.

Although the average values of HI (sum of TQH for Pb, Cd, and As) in the population from 2 to 85 years in the scenarios of minimum, average, and maximum milk consumption were 0.26, 0.34, and 0.45, respectively, the HI values were greater than 1 in children under 7, 9, and 11 years of age for the minimum, average, and maximum milk consumption, respectively; that is to say, the greater the consumption, the greater the potential risk of exposure to these toxic elements. In the adults, the HI were <1 (Figure 6). These results are interesting because they provide information differentiated by age and daily intake level of milk produced on a farm that adjoins both a mini mineral concentrator and an old municipal landfill, which for decades has polluted the air, water, soil, and the products obtained through this production system.

## 4. Discussion

### 4.1. Pb, Cd, and As Concentrations in the Study Samples

#### 4.1.1. Pb, Cd, and As Concentrations in the Soil

The concentration of Pb in the soil exceeded the MPL for agricultural soils in Peru by nine times [32] due to many years of exposure to emissions from the adjacent mini concentrator, which has been operating since 1965, and to their proximity to the municipal solid waste landfill. These results are in the range reported in different regions of the world, whose variations reflect geographical areas, the level of anthropogenic activity, and the different regulations that ultimately affect environmental conditions [6,29,55,56].

The concentration of Cd in the soil exceeded the MPL by five times, and that of the grass represented 25% of the MPL. The primary source of these contents would be organophosphate residues used in agriculture in the area and from the farm itself, in addition to emissions from the mini mineral concentrator and from the former organic waste landfill and irrigation water from the Mantaro River, known for its high contamination both in source in Lake Junín and during its course [55]. It is essential to consider that phosphate rock contains up to 300 mg/kg. Cd limits in phosphate rock-based fertilizers in Peru are 2.0–186 mg/kg, while in Mexico and Brazil, they are 4.0 and 8.0 mg/kg, respectively [57], demonstrating that the Peruvian standard is highly permissible.

Regarding As, its mean concentrations in the soil was below the MPL, and it was 28 times higher than those reported in other less contaminated regions of the world [56], which indicates that the study area has a high level of contamination. The As of the pastures was below the MPL. Our results are related to reports from Brazil, indicating that contamination by As occurs mainly in regions with intensive mining activity [58].

In a study carried out in southern Peru (Tiquillaca-Puno) in a mining waste dumping area, the average contents of Pb, Cd, and As in the soil were 276.74, 1.45, and 5.35 mg/kg, respectively, and these values are lower than those of our study [34]. In another study carried out in the Peruvian Amazon [59] in soils abandoned by gold mining, 12.6, 0.79, and 6.67 mg/kg were reported for Pb, Cd, and As, respectively, with values of Pb and Cd were much lower than those of our study, but similar in As.

#### 4.1.2. Pb, Cd, and As Concentrations in Multi-Species Pasture

The concentrations of Pb, Cd, and As in the grasses of multiple species represented 77, 25, and 3% of the MPL (Table 1); the mean Pb content was even higher than that found in an area 20 km from the La Oroya Metallurgical Complex, a city included among the ten most polluted cities on the planet [55]. Mean Cd and As concentrations in the multi-species pasture were below those reported in other world regions, such as China, where cattle feed contained 2.31 and 1.38 mg/kg, respectively [60]; similarly, in soils in Dong Mai, Hung Yen-Vietnam, where natural and cultivated plants are cultivated near a Pb recycling zone, lower Pb and Cd contents (5.4–26.8 and 0.71–1.67 mg/kg) but higher As were also reported (370–47,400 mg/kg) [17].

In the exotic grasslands of southern New Zealand, high levels of Pb and Cd have been found, indicating that the application of phosphorus-based fertilizers and proximity to urban centers and busy roads are the principal sources of heavy metals [29]. These heavy metals are deposited and accumulated in the upper layer of the soil and are absorbed by the pastures, passing into the food chain, and these results are in line with different studies [61,62].

Comparing our results with investigations carried out in Peru, the Pb content in cultivated pastures used for cattle feed was four times higher than that reported in the Mantaro Valley, which was 5.8 mg/kg [63], and this is expected because in this case, the mini mineral concentrator mainly emits Pb into the environment. The Cd content in this study was 11 times lower than that reported in another area of the Mantaro Valley, where uncontrolled use of phosphorous agrochemicals used in intensive agriculture is practiced in that area. In any case, the proximity of the farm in our study to a mini mineral concentrator and a former municipal organic waste dump, as well as the use of phosphorous fertilizer in pasture production, would be the main determinants for the presence of Pb. The presence Cd and As in the soils and pastures is detrimental, and remedial actions must be taken for farm soils.

#### 4.1.3. Pb, Cd, and As Concentrations in Milk

The mean contents of Pb, Cd, and As in raw cow’s milk (Table 1) exceeded the MPL by 3.1, 4.7, and 2.1 times [46,64,65,66], a worrying result because the central highlands of Peru are recognized as one of the food pantries of the capital Lima; hence, this milk, which is consumed directly or collected for industrialization, would not be suitable for human consumption, and would represent a potential health risk.

At a global level, in a review of studies carried out between 2010 and 2020 on the concentration of heavy metals in fresh milk, a Pb content of up to 60 mg/kg was reported in a granite and gneiss mining area in India, indicating that the different concentrations depend on the environmental, as well as anthropogenic and regulatory control conditions of each area [67]. Thus, the Pb MPLs in cow’s milk also vary in Germany and the Netherlands, Turkey, and Russia, being 0.05, 0.02, and 0.10 mg/kg, respectively [68], while most Latin American countries do not have these standards.

As the contents of Pb, Cd, and As in the raw cow’s milk evaluated in this study exceeded the MPL, this food is unsuitable for human consumption and represents a potential long-term health risk, an item discussed later. The concentrations of these three toxic elements followed the descending order Pb > As > Cd, representing 58.5, 28.3, and 13.2% of the total.

Among the studies that evaluated the content of Pb, Cd, or As in raw cow’s milk which showed results similar to ours, we will mention them in chronological order, beginning with the one from Calabria, Italy, reporting an average As content of 0.038 mg/kg of raw milk. [69]. A study in an industrial region of Zagreb, Croatia, reported concentrations of Pb and Cd between 0.02 and 0.06 and 0.003 and 0.006 mg/L [70]. In Croatia itself, on farms in rural areas in the north and south of the country, mean concentrations of As, Cd, and Pb of 0.019 and 0.044, 0.002, and 0.003, and 0.059 and 0.036 mg/L [71] are reported. In Beni-Suef, Egypt, concentrations of Pb and Cd between 0.044 and 0.751 and 0.008 and 0.179 mg/kg were reported [72]. In Arak, Iran, residual amounts of As between 0.015 and 0.026 mg/kg were found [73]. In Mansoura, Egypt, a Pb content of 0.07 mg/kg [74] was found, and in Kazakhstan, Cd concentrations of 0.01 and 0.02 mg/L were reported in summer and autumn, respectively [56]. All these results would respond to contamination conditions similar to those in our study, where the smelting activity would be the main emitter of Pb, Cd, and As.

Among the investigations that report lower values than ours are those from Calabria, Italy, with a Pb content of 0.0013 mg/kg of raw milk [66]; in Córdoba, Argentina, with an As content in raw milk in the range of 0.0003–0.0105 mg/kg [75]; in farms of municipalities in northeastern Iran, with mean levels of 0.012 and 0.0003 mg/kg for Pb and Cd [76]; in industrial regions of Iran, with mean Pb and Cd levels of 0.014 and 0.001 mg/kg [77]; in milk collected between 2010 and 2014 in small rural dairy farms in Croatia, with mean Pb concentrations between 0.011 (2013) and 0.012 (2014) mg/kg [78]; in industrial cities of Isfahan and Ahvaz, Iran, with mean Pb and Cd levels of 0.014.0 and 0.001 mg/L [77]; in milk samples marketed in Istanbul, with mean As, Cd, and Pb concentrations of 0.0026, 0.0001, and 0.002 mg/kg [79]; and in Slovakia, with mean contents of As < 0.03 mg/kg, Pb < 0.10 mg/kg, and Cd < 0.004 mg/kg [80]. All these low contents of Pb, Cd, and As in raw cow’s milk are indicative that its direct use or its use in the transformation of dairy products does not pose risks to the health of consumers, being safe for human consumption.

Among the studies that report values higher than ours, and which exceed the Codex Alimentarius standards, were farms in India near a lead–zinc smelter and a steel manufacturing plant with mean concentrations of Pb and Cd of 0.85 and 0.23 mg/L [81]. In Brasília, Brazil, they found average amounts of Pb of 0.084 mg/kg [82]; in El-Qaliubiya-Egypt, Pb, and Cd of 4.40 and 0.29 mg/kg were found [80]; and in Mansoura, Egypt, 0.104 mg/kg of Cd were found [83]. In areas close to leather processing plants in China, As and Pb contents of 0.0043 and 0.0029 mg/L are reported [84].

These results clearly demonstrate that the level of heavy metals in milk is associated with different factors such as the breed of cattle, the type of feed, the production system, the use of agrochemicals, the proximity to mining–metallurgical activities of milk production sites, and the inadequate disposal of solid waste, which determines the high levels of heavy metals in water, soil, and pastures entering the food chain. These findings agree with other studies [85].

### 4.2. Week Intake (WI), Target Hazard Quotient (TQH) and Hazard Index (HI) to Pb, Cd, and As

In general, the WIs for Pb, Cd, and As increased as milk intake increased up to 19 years and then decreased until 85 years. In the case of Pb and Cd, no intake level for age exceeded the corresponding TWIs [43], while the WIs for As at all consumption levels and ages exceeded the TWIs established by the USEPA [44].

In children aged 2 and 3 years, with minimum, average, and maximum milk intake, the WIs for As were 21, 27 and 32 times the TWI, decreasing to 7 times the TWI in people aged 84 to 85 years (Table 3). These results indicate that the principal contaminant of the milk produced in the environmental conditions of the study site is As, which would have as sources of exposure the emissions and leachates from the mineral mini-concentrator and those coming from the old municipal landfill. This milk, due to its high As content, turns out to be unfit for human consumption.

The WI values of Pb and Cd estimated in this study were similar to those reported in other regions of the world, such as Alexandria, West Delta, Egypt, where the WI represented less than 5% of the TWI for both metals, determining TQHs < 1 [86]; those of Tehran-Iran, which also reports TQH < 1 for Pb and Cd [87]; those of China, where negligible risks are reported from the consumption of milk with Pb, Cd, and As [8,88]; and as in our study, the risks are increasing in young children because they consume more milk than adults per kilogram of body weight.

In China, in places close to leather processing plants, in people from 3 to 69 years of age, the TQH followed a descending order of As > Pb > Cr > Cd, with values of 0.007–0.044, 0.003–0.022, 0.002–0.012, and 0.0017–0.005, respectively, with HI values (0.012–0.083) well below the threshold of 1, and therefore no representing a health risk [84]. In our case, for that same age group, the TQH for mean milk consumption also followed an ascending order of As > Pb > Cd, with values of 0.12–1.41, 0.03–0.35, 0.03–0.30, respectively, with HI values (0.18–2.06) higher than the threshold of 1 in children from 2 to 9 years of age, which represents a potential risk to the health of these infants.

In Guelma, Algeria, THQs for Pb and Cd suggest a potential risk to infants, with HI > 1; the contributions of the metals followed a decreasing order of Pb, Cr, Cd, Ni, Zn, Cu, and Fe with values of 68.19%, 15.39%, 6.91%, 4.94%, 3.42%, 0.88%, and 0.28%, respectively [45]. In our study, for 2-year-old babies with average milk intake, the As, Pb, and Cd represented 68.44%, 17.00%, and 14.56%, respectively, with As being the metalloid with the highest risk.

A review of research carried out between 2010 and 2020 reported the non-cancer risk of Pb for milk consumers in 10 out of 70 regions exposed to some potential health risk, with the highest TQH for Pb 35.2 in milk produced in Tirupati province, India, an area heavily polluted by lead–acid battery factories [67]. THQ >1 is also recorded in the Peshawar and Faisalabad provinces in Pakistan; in the Şakirbey, Yeniçiftlik, and Gümüşçay provinces in Turkey; in Nitra province, Slovakia; in the Tokh-El-Qaliubiya industrial zone, Egypt; and in the Mumbai region of India [67]. Regarding Cd, TQH values in 6 of 59 regions were >1. Yhe highest value (TQH = 24) was also recorded in Tirupati province, India. TQH > 1 is reported in Sakirbey, Yeniçiftlik, and Gümüşçay provinces in Turkey and Peshawar and Sargodha provinces in Pakistan [67].

Our results, and those indicated in this discussion, reveal that a continuous intake of contaminated milk, even with low levels of metals and toxic metalloids, would represent health risks for consumers. Industrial and agricultural activities go hand in hand with an increase in the concentrations of metals in the air, water, and soil, which are absorbed by plants and accumulate in their tissues, passing, in turn, to milk from animals that consume contaminated plants, incorporating heavy metals into the food chain [89]. In the world, there are differences in the levels of metals in cow’s milk, depending on the countries, specific areas, and regulations [4]. For example, in Ecuador, the Technical Standard No. NTE 0009:2008 for As considers 0.015 mg/kg as MPL [90], and in our study, twice that value was found. The People’s Republic of China Standard establishes 0.10 mg/kg as MPL. The official Mexican standard NOM-184-SSA1-2002 considers 0.2 mg/kg as MPL, and MercoSur-Res. No. 12/11 suggests 0.05 mg/kg [90]. This variability in the MPL deserves further analysis in order to establish with greater precision the contents of heavy metals and other toxic substances in staple foods, especially those used in infant feedings such as milk and its derivatives.

Finally, our findings indicate that the milk produced under the conditions of the study area would not be environmentally safe or suitable for consumption by the Peruvian population due to its Pb, Cd, and As content, and we suggest taking quality control measures with respect to milk production, the farm, heavy metals monitoring, and As in irrigation water and soil, and to carry out good agricultural practices. Studies such as this provide the information necessary to guide the different sustainable initiatives for the control and evaluation of environmental contamination and, above all, the mapping of productive areas of high risk to human health. Studies should continue with a greater number of heavy metals and regions of Peru in order to obtain scientific evidence that supports sustainable development proposals.

### 4.3. Implications of the Intake of Pb, Cd, and As in Mental Health

Pb, even at low levels, is known to impair brain development and neurobehavior; however, it represents a small fraction of the neurotoxic (NT) exposome. Simultaneous exposure of Pb, Cd, and As has severe effects on the CNS, causes cognitive impairment, lowers behavioral test results, and leads to abnormal social behavior and depressive disorders [91,92,93,94]. Multiple exposures to NT mixtures in children diagnosed with autism spectrum disorders and attention deficit hyperactivity disorders suggest a strong Pb-associated effect [95]. Generalized anxiety disorders, panic disorder/agoraphobia, social anxiety disorder, suicide, and other psychological conditions are burdensome for communities [96,97,98,99]. Clinical studies have shown that patients with depressive disorders and panic attacks had an excess concentration of certain metals in their system such as Cd and Pb [100].

The most toxic and ubiquitous class 1 elements (Cd, As, and Pb) should be considered in risk assessment [101]; however, there are no studies evaluating the risk of these metals for psychological problems, so it is necessary to monitor the levels of heavy metals and metalloids consumed in food. Another study in preschoolers determined the levels of heavy metals and neurotransmitters in the blood, together with neurobehavioral scores, observing that the increase in Pb was associated with decreases in glutamic acid, glycine, gamma-aminobutyric acid, and serine, which increased hyperactivity index and anxiety [102]. Other work has shown associations between the level of As in the blood of pregnant women with a low level of brain-derived neurotrophic factor, which is associated with maternal depressive disorder and neurodevelopment of the newborn [93].

Studies carried out in Peru on mental health and its relationship with exposure to Pb, Cd, and As in children and adults surrounding the Las Bambas mining project in Apurímac, before the exploitation phase evaluating psychomotor development in children < 3 years, with the TEPSI test, reported a 12.5% more risk in psychomotor development [103]. The IQ with attained via Stanford-Binet test in children aged 3–12 years reports cases of mild mental disability (2.1%) and borderline mental disability (3.1%) [103]. As to levels of anxiety and depression in the population > 12 years, there are report of 34.3% suffering anxiety and 17.5% depression, and more emissions of heavy metals would aggravate this epidemiological situation [103]. In Callao, when studying the intellectual levels and anxiety in children chronically intoxicated with Pb, categorizing the children into non-intoxicated (<10 mg/dL) and intoxicated (>10 mg/dL), they report a lower IQ and a higher level of anxiety among the intoxicated [104].

In summary, simultaneous exposure to Pb, Cd, and As during pregnancy and lactation can affect neurodevelopment in the first years of life, which can lead to autism spectrum disorders and attention deficit hyperactivity disorders, anxiety disorders, and depression; however, there are not enough studies on the risks of these contaminants for mental health, making it necessary to monitor the levels of these toxins in food and to implement public health policies to protect the fetus and young children from exposure to these contaminants.

The results of this study provide evidence of the potential risk from the consumption of milk contaminated with Pb, Cd, and As in the Peruvian population, with the highest risk being to the child population. However, more research is needed on the levels of heavy metals in a greater number of milk samples form the different dairy basins of the country, both to examine this problem from the clinical–epidemiological point of view and to identify the possible causes of milk contamination.

## 5. Conclusions

In this investigation, the levels of Pb, Cd, and As in the milk produced in the vicinity of a mini mineral concentrator and a former municipal landfill, and which were also influenced by the use of phosphate rock fertilizers, exceeded the limit values established internationally. The Pb, Cd, and As presence in the milk from the contaminated area could be due to the water, soil, and grass containing high heavy metals levels.

The health risk assessment indices, which include the TQH and HI, indicated a potential health risk to Peruvian children under 7, 9, and 11 years of age in low, medium, and high consumption scenarios of bovine milk contaminated with Pb, Cd, and As which is produced in an area adjacent to a mini mineral concentrator plant and a municipal solid waste dump, with the As being the element posing the highest risk.

This study allows us to broaden the scientific basis for the safe and innocuous production of bovine milk in the basin of the central Andes of Peru, for the establishment of a continuous monitoring program for toxic metals in raw milk, and for the development of national standards by which to recover agricultural land contaminated with heavy metals and to guarantee the production of fodder and milk with a minimum content of heavy metals and thus a low risk for human consumption.

## Figures and Tables

**Figure 1 toxics-11-00809-f001:**
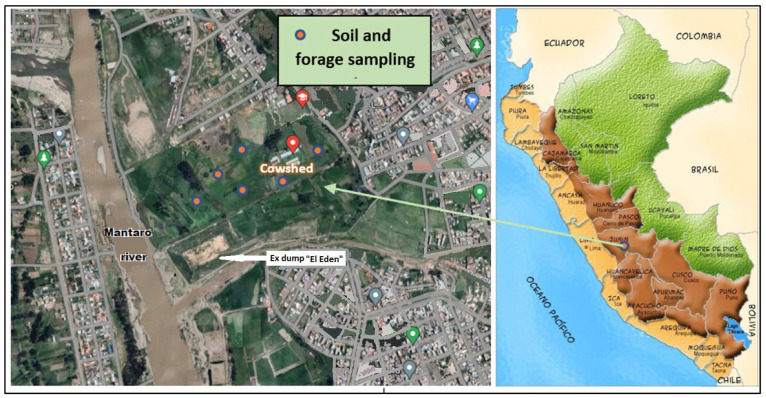
Study area map and sampled location [25,26].

**Figure 2 toxics-11-00809-f002:**
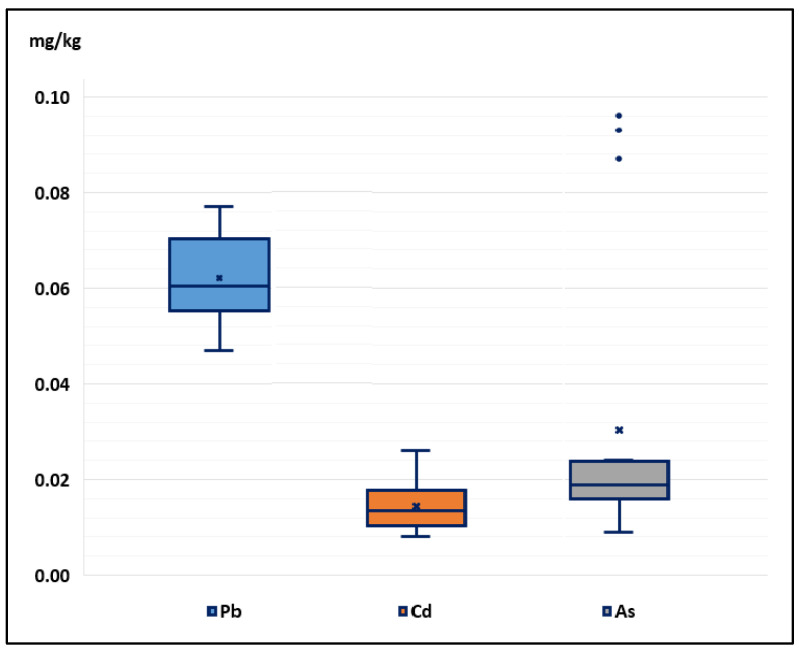
Box and whisker plot representing the dataset of Pb, Cd, and As content in milk samples.

**Figure 3 toxics-11-00809-f003:**
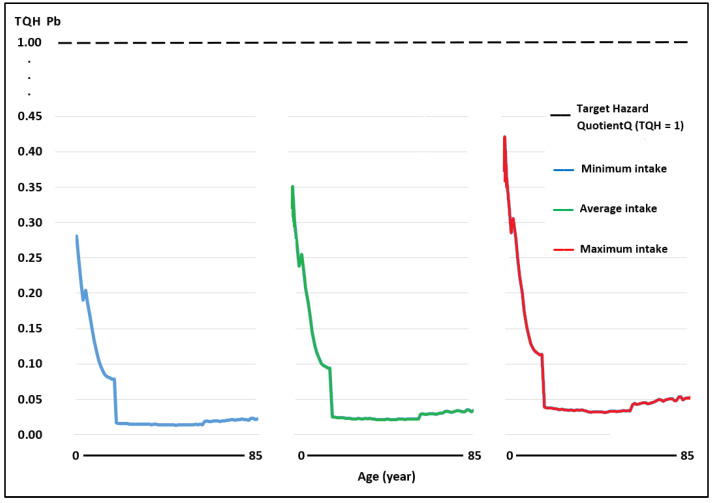
Target risk quotient (TQH) for Pb in milk consumption in people aged 2–85 years at minimum, medium, and maximum exposure.

**Figure 4 toxics-11-00809-f004:**
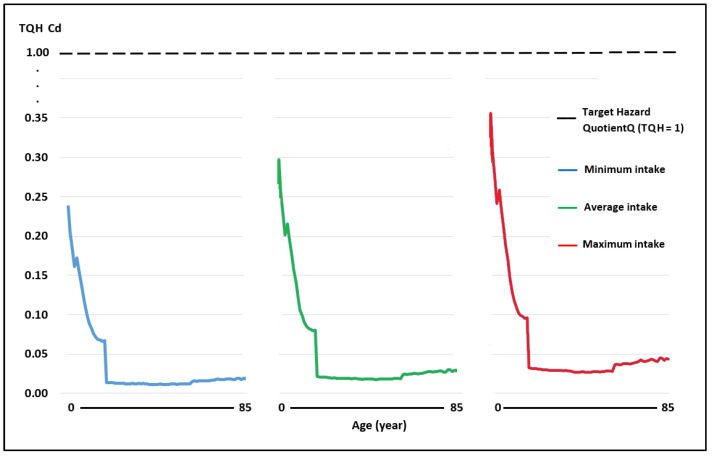
Target risk quotient (TQH) for Cd in milk consumption in people aged 2–85 years at minimum, medium, and maximum exposure.

**Figure 5 toxics-11-00809-f005:**
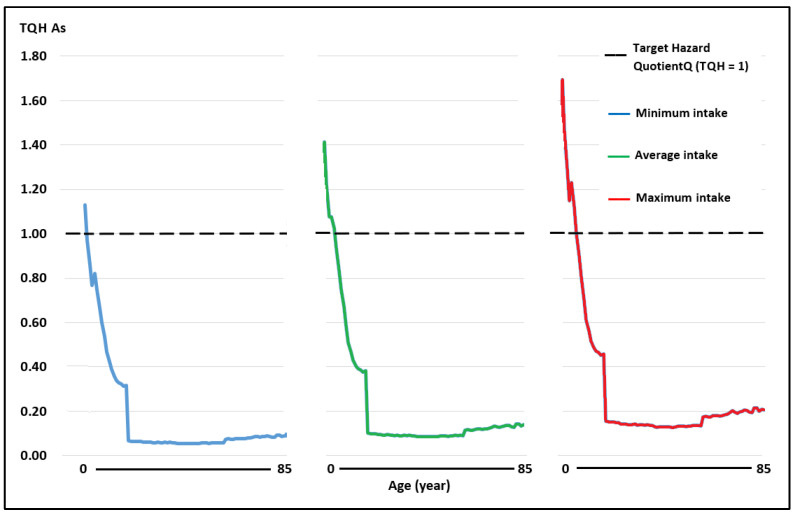
Target risk quotient (TQH) for As for milk consumption in people aged 2–85 years at minimum, medium, and maximum exposure.

**Figure 6 toxics-11-00809-f006:**
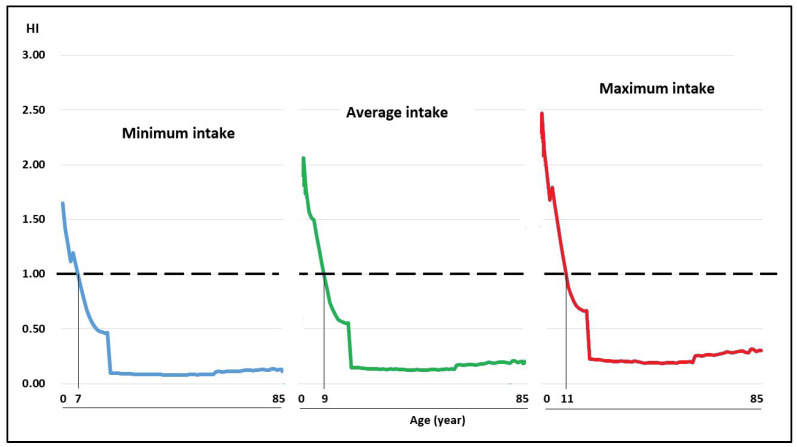
Hazard index (HI) for minimum, medium, and maximum milk intake contaminated with Pb, Cd, and As in the Peruvian population from 2 to 85 years of age.

**Table 1 toxics-11-00809-t001:** Pb, Cd, and As concentration in the soil, multi-species cultivated pasture shoots, and milk (mg/kg), transfer, and bioaccumulation percentages in a central area of the Peruvian Andes.

	Pb			Cd			As		
	Soil	Sproud	Milk	Soil	Sproud	Milk	Soil	Sproud	Milk
Mean	652.35 a	23.17 b	0.062 c	7.09 a	0.250 b	0.014 c	6.29 a	0.063 b	0.030 c
SD	192.50	10.02	0.008	5.31	0.57	0.005	4.98	0.09	0.028
Min	339.69	9.32	0.047	1.73	0.002	0.008	3.49	0.01	0.009
Max	879.00	34.00	0.077	19.59	1.40	0.026	16.41	0.34	0.096
MPL	70	30	0.02	1.4	1.0	0.003	12	2.0	0.014
T, %	100.00	3.55	0.27	100.00	3.52	5.60	100.00	1.00	47.62
B, %	100.00		0.01	100.00		0.20	100.00		0.48

a,b,c, Average values per item with different letters vary statistically (*p* < 0.01). SD: standard deviation; Min: minimum; Max: maximum; MPL: maximum permissible limit; T, %: percentage transfer from soil to pasture and from pasture to milk; B, %: percent bioaccumulation (from soil to milk).

**Table 2 toxics-11-00809-t002:** Weekly intake (WI) for minimum, medium, and maximum milk intake contaminated with Pb, Cd, and As in the Peruvian population from 2 to 85 years of age (*n* = 62,600).

Age (Year)	Minimum Milk Intake			Medium Milk Intake			Maximum Milk Intake			TWI Pb	TWI Cd	TWI As
	WI Pb	WI Cd	WI As	WI Pb	WI Cd	WI As	WI Pb	WI Cd	WI As			
2	174.16	40.32	85.12	217.70	50.40	106.40	261.24	60.48	127.68	310	72	4
3	174.16	40.32	85.12	217.70	50.40	106.40	261.24	60.48	127.68	360	84	4
4	174.16	40.32	85.12	217.70	50.40	106.40	261.24	60.48	127.68	403	93	5
5	174.16	40.32	85.12	217.70	50.40	106.40	261.24	60.48	127.68	448	104	5
6	208.99	48.38	102.14	261.24	60.48	127.68	313.49	72.58	153.22	503	117	6
7	208.99	48.38	102.14	261.24	60.48	127.68	313.49	72.58	153.22	550	128	7
8	208.99	48.38	102.14	261.24	60.48	127.68	313.49	72.58	153.22	615	143	7
9	208.99	48.38	102.14	261.24	60.48	127.68	313.49	72.58	153.22	670	155	8
10	208.99	48.38	102.14	261.24	60.48	127.68	313.49	72.58	153.22	740	172	9
11	208.99	48.38	102.14	261.24	60.48	127.68	313.49	72.58	153.22	818	190	10
12	217.70	50.40	106.40	261.24	60.48	127.68	313.49	72.58	153.22	910	211	11
:												
81	52.25	12.10	25.54	80.55	18.65	39.37	121.04	28.02	59.16	1478	343	18
82	52.25	12.10	25.54	80.55	18.65	39.37	121.04	28.02	59.16	1475	342	18
83	52.25	12.10	25.54	80.55	18.65	39.37	121.04	28.02	59.16	1368	317	16
84	52.25	12.10	25.54	80.55	18.65	39.37	121.04	28.02	59.16	1415	328	17
85	52.25	12.10	25.54	80.55	18.65	39.37	121.04	28.02	59.16	1390	322	17

WI: weekly intake for Pb, Cd, and As; TWI: tolerable weekly intake for Pb, Cd, and As.

**Table 3 toxics-11-00809-t003:** Target hazard coefficient (TQH) and hazard index (HI) for the minimum, medium, and maximum consumption of milk contaminated with Pb, Cd, and As in a central area of the Peruvian Andes.

Age (Year)	Milk Intake, kg/d	Minimum Milk Intake				Milk Intake, kg/d	Medium Milk Intake				Milk Intake, kg/d	Maximum Milk Intake			
		TQH Pb	TQH Cd	TQH As	HI		TQH Pb	TQH Cd	TQH As	HI		TQH Pb	TQH Cd	TQH As	HI
2	0.40	0.28	0.24	**1.13**	**1.65**	0.50	0.35	0.30	**1.41**	**2.06**	0.60	0.42	0.36	**1.69**	**2.47**
3	0.40	0.24	0.20	0.97	**1.42**	0.50	0.30	0.26	**1.22**	**1.77**	0.60	0.36	0.31	**1.46**	**2.13**
4	0.40	0.21	0.18	0.86	**1.25**	0.50	0.27	0.23	**1.08**	**1.57**	0.60	0.32	0.27	**1.29**	**1.88**
5	0.40	0.19	0.16	0.77	**1.12**	0.50	0.24	0.20	**1.05**	**1.49**	0.60	0.29	0.24	**1.15**	**1.68**
6	0.48	0.20	0.17	0.82	**1.20**	0.60	0.25	0.22	**1.03**	**1.50**	0.72	0.31	0.26	**1.23**	**1.80**
7	0.48	0.19	0.16	0.75	**1.09**	0.60	0.23	0.20	0.93	**1.36**	0.72	0.28	0.24	**1.12**	**1.64**
8	0.48	0.17	0.14	0.67	0.97	0.60	0.21	0.18	0.83	**1.22**	0.72	0.25	0.21	**1.00**	**1.46**
9	0.48	0.15	0.13	0.60	0.87	0.60	0.19	0.16	0.75	**1.09**	0.72	0.22	0.19	0.90	**1.31**
10	0.48	0.13	0.11	0.54	0.78	0.60	0.17	0.14	0.67	0.98	0.72	0.20	0.17	0.81	**1.17**
11	0.48	0.12	0.10	0.47	0.68	0.60	0.14	0.12	0.58	0.85	0.72	0.17	0.15	0.70	**1.02**
12	0.50	0.11	0.09	0.42	0.62	0.60	0.13	0.11	0.51	0.74	0.72	0.15	0.13	0.61	0.89
:															
81	0.12	0.02	0.02	0.09	0.13	0.19	0.04	0.03	0.14	0.21	0.28	0.05	0.05	0.21	0.31
82	0.12	0.02	0.02	0.09	0.13	0.19	0.04	0.03	0.14	0.21	0.28	0.05	0.05	0.21	0.31
83	0.12	0.02	0.02	0.09	0.13	0.19	0.03	0.03	0.13	0.19	0.28	0.05	0.04	0.20	0.29
84	0.12	0.02	0.02	0.09	0.13	0.19	0.03	0.03	0.14	0.20	0.28	0.05	0.04	0.21	0.31
85	0.12	0.02	0.02	0.09	0.13	0.19	0.03	0.03	0.14	0.20	0.28	0.05	0.04	0.21	0.30

TQH: Target hazard coefficient for Pb, Cd, and As. HI: hazard index for Pb, Cd, and As.

## Data Availability

The research data is found in the manuscript.

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
