# Peer review of "Lead, Cadmium, and Arsenic in Raw Cow’s Milk in a Central Andean Area and Risks for the Peruvian Populations"

_toxics, 2023, doi:10.3390/toxics11100809_

Round 1

Reviewer 1 Report

General comments: The manuscript is interesting and fits well in the scope of Toxics. However, I believe that the Authors have not explained why they analyzed soil and feed samples in addition to milk. Such analyses would make sense if, for example, different regions were compared or if the transmission of metals in the soil-plant-milk system was studied. In the case of the results presented by the Authors, there is no need to present the content of toxic metals in the soil and feed. Additionally, there are many typos and punctuation errors in the text - the text should be proofread.

Specific comments:

1. Abstract: well-written

2. Keywords: should be added „cadmium” and „lead”, should be removed „heavy metals”

3. Introduction:

- line 70 „CDC”: explain the abbreviation

- a disproportionate amount of space is devoted to As compared to Cd and Pb, which are also toxic

- aim of the work: It should be added that the content of Cd, Pb, and As in the soil and grass was also determined (do the authors mean multi-species pasture green fodder? If so, this should be indicated in the entire manuscript).

4. Materials and methods:

- please specify the determination parameters for Cd, Pb and As

5. Results

- 3.2. Ingesta Diaria Estimada (WI) y Coeficiente de riesgo dietario (DRC): please translate to English

- line 293 and 306: „… consumption scenarios. , correspondingly” – please remove the dot6.   

6. Disussion: The Discussion is chaotic, maybe it would be better to divide it into age groups (pre-school, school, youth, adults, and the elderly) and discuss the results in this way? Now the discussion is difficult for the reader to follow.

7. Conclussions: appropriate

8. References: appropriate, updated, to be formatted in accordance with editorial requirements

9. Tables: appropriate

10. Figures: appropriate

Author Response

Response to Reviewer 1 Comments

General comments: The manuscript is interesting and fits well in the scope of Toxics. However, I believe that the Authors have not explained why they analyzed soil and feed samples in addition to milk. Such analyses would make sense if, for example, different regions were compared or if the transmission of metals in the soil-plant-milk system was studied. In the case of the results presented by the Authors, there is no need to present the content of toxic metals in the soil and feed. Additionally, there are many typos and punctuation errors in the text - the text should be proofread.

Response to general comments: We have improved the manuscript based on the final comments of the reviewers, to whom we sincerely thank them, as their comments have improved the quality of the article.

The research design has been better explained, including the determination of Pb, Cd, and As in soils and pastures as an objective. The map better supports the study design.

Considering the general comment that the research has not studied the transmission of heavy metals in the soil-plant-milk system and there would be no need to present them, we believe that these results complement the work, and this aspect has also been included as an objective of the study.

Typos and punctuation errors have been fixed.

Specific comments:

Point 1. Abstract: well-written

Response 1: We appreciate this review which strengthens our writing skills.

Point 2. Keywords: should be added „cadmium” and „lead”, should be removed „heavy metals”

Response 2: We appreciate this observation that will allow better identification of the article by search engines.

Point 3. Introduction:

- line 70 „CDC”: explain the abbreviation

Response: The meaning of the abbreviation has been included at length and it has been placed in parentheses.

- a disproportionate amount of space is devoted to As compared to Cd and Pb, which are also toxic

Response: The paragraph has been rewritten, removing the information that is not specific to the present study and we believe that there is now a proportion between what was stated for the three elements studied.

- aim of the work: It should be added that the content of Cd, Pb, and As in the soil and grass was also determined (do the authors mean multi-species pasture green fodder? If so, this should be indicated in the entire manuscript).

Response: We have added this objective in the text. Also, the term multispecies green grass has been considered throughout the text.

Point 4. Materials and methods:

- please specify the determination parameters for Cd, Pb and As

Response 4: In Table 1, the parameters for the determination of Pb, Cd, and As in the soil, grass, and milk have been added, including minimum and maximum.

Point 5. Results

- 3.2. Ingesta Diaria Estimada (WI) y Coeficiente de riesgo dietario (DRC): please translate to English

Response: It was an inadvertent error and these terms have been properly written in English. For a better understanding of the results, Table 2 on weekly intake has been modified, and Table 3 on the results of TQH and HI has been included.

- line 293 and 306: „… consumption scenarios. , correspondingly” – please remove the dot6. 

Response: This punctuation bug has been removed.

Point 6. Disussion: The Discussion is chaotic, maybe it would be better to divide it into age groups (pre-school, school, youth, adults, and the elderly) and discuss the results in this way? Now the discussion is difficult for the reader to follow.

Response 6: We have synthesized and formatted the discussion better. Regarding risks, we have rewritten item 4.2 on WI, TQH, and HI, and we have included tables of results.

Point 7. Conclussions: appropriate

Response 7: We appreciate the opinion.

Point 8. References: appropriate, updated, to be formatted in accordance with editorial requirements

Response 8: We appreciate the opinion.

Point 9. Tables: appropriate

Response 9: We appreciate the opinion.

Point 10. Figures: appropriate

Response 10: We appreciate the opinion.

Jorge Castro Bedriñana

Reviewer 2 Report

This study sheds light on a critical concern, and I believe that your findings will contribute significantly to the ongoing efforts in enhancing milk safety and protecting the health of the Peruvian population. I recommend emphasizing the practical implications of your research for policy formulation and awareness campaigns in the final article. Additionally, consider addressing potential limitations or areas for further research in order to provide a comprehensive perspective.

You need more improve the Quality of English language for better understanding and interesting for reader 

Author Response

Response to Reviewer 2 Comments

Point 1. This study sheds light on a critical concern, and I believe that your findings will contribute significantly to the ongoing efforts in enhancing milk safety and protecting the health of the Peruvian population. I recommend emphasizing the practical implications of your research for policy formulation and awareness campaigns in the final article. Additionally, consider addressing potential limitations or areas for further research in order to provide a comprehensive perspective.

Response 1. We appreciate the recommendation. In the last paragraph of 4.2, we have included practical suggestions that include awareness raising; Likewise, we have pointed out the need to continue studies with a greater number of toxic metals in the most polluted regions of the country.

Point 2. You need more improve the Quality of the English language for better understanding and interest for reader. 

Response 2. We appreciate the recommendation. A general revision of the English has been made to facilitate its understanding. Likewise, the conclusions have been improved.

Jorge Castro Bedriñana

Reviewer 3 Report

                                                                                                         Dated: 06-08-2023

MS_Toxics_2558523

Title: Lead, Cadmium, and Arsenic: Levels in Raw Cow’s Milk in a Central Andean Area and Risks for the Peruvian Populations

Dear Sir,

The work is interesting and has a scientific aspect, but there are some/minor problems with the manuscript. The manuscript is acceptable for publication after minor revisions. Please see the list of comments.

Remark: Accept after minor revision

List of comments

1.     The abstract should be revised more critically.

2.     Please check the journal guidelines for the number of “Keywords”.

3.     Line number 70; CDC give the full name also.

4.     2.1. Study Period and Location: Provide geocoordinate of the study area.

5.     Dietary Risk Coefficient (DRC): Give the citation/reference.

6.     Materials and methods section; Hazard Index (HI): It should be in a separate point/heading with proper citation/reference.

7.     The conclusion section should also be revised more critically.

                                                                                                            Dated: 06-08-2023

MS_Toxics_2558523

Title: Lead, Cadmium, and Arsenic: Levels in Raw Cow’s Milk in a Central Andean Area and Risks for the Peruvian Populations

Dear Sir,

The work is interesting and has a scientific aspect, but there are some/minor problems with the manuscript. The manuscript is acceptable for publication after minor revisions. Please see the list of comments.

Remark: Accept after minor revision

List of comments

1.     The abstract should be revised more critically.

2.     Please check the journal guidelines for the number of “Keywords”.

3.     Line number 70; CDC give the full name also.

4.     2.1. Study Period and Location: Provide geocoordinate of the study area.

5.     Dietary Risk Coefficient (DRC): Give the citation/reference.

6.     Materials and methods section; Hazard Index (HI): It should be in a separate point/heading with proper citation/reference.

7.     The conclusion section should also be revised more critically.

Author Response

Response to Reviewer 3 Comments

Point 1. The abstract should be revised more critically.

Response 1. We appreciate the recommendation. The summary has been revised, and some refinements have been made.

Point 2.  Please check the journal guidelines for the number of “Keywords”.

Response 2. We have consulted the guidelines, and when loading the keywords range of 3 to 10 keywords that we have considered is indicated. 

Point 3. Line number 70; CDC give the full name also.

Response 3. We appreciate the recommendation. The full name of the CDC has been included, indicating the initials in parentheses. 

Point 4. 2.1. Study Period and Location: Provide geocoordinate of the study area.

Response 4. Thanks for the important observation. We have included the geographic coordinates of the study site. 

Point 5. Dietary Risk Coefficient (DRC): Give the citation/reference.

Response 5. To provide clarity and avoid confusion, what corresponds to DRC has been removed, to emphasize TQH and HI, the reason for the study. 

Point 6. Materials and methods section; Hazard Index (HI): It should be in a separate point/heading with proper citation/reference.

Response 6. In the manuscript, we have included the danger index with appropriate references as item 2.6. 

Point 7. The conclusion section should also be revised more critically.

Response 7. The conclusions have been revised and improved. We also inform you that English has been improved. 

Jorge Castro Bedriñana

Reviewer 4 Report

1.The introduction section is too long and should be appropriately deleted. 2.The grammar of the entire text needs to be extensively revised.
3.The map in the article needs to indicate its source.
4.The references need to be adjusted according to the journal format.
5.The abstract section should reflect innovation and parameters.

Extensive editing of English language required

Author Response

Response to Reviewer 4 Comments

Point 1. The introduction section is too long and should be appropriately deleted.

Response 1. Thanks for the recommendation. In the introduction, extensive information on As has been removed, and the Pb, Cd, and As information has been treated proportionately.

Point 2. The grammar of the entire text needs to be extensively revised.
Response 2. Thanks for the observation. Grammar has been checked throughout the manuscript.

Point 3. The map in the article needs to indicate its source.

Response 3. We have included the initial source of the map on which the soil and pasture sampling points have been included.

Point 4. The references need to be adjusted according to the journal format.

Response 4. We appreciate this contribution. We have adjusted the references according to the format of the journal.

Point 5. The abstract section should reflect innovation and parameters.

Response 5. We appreciate this input and have adjusted the summary based on your recommendation.

Point 6. Comments on the Quality of the English Language: Extensive editing of the English language required

Response 6. English has been improved throughout the text

Jorge Castro Bedriñana

Round 2

Reviewer 1 Report

The manuscript is very interesting and fits in the scope of the Toxics journal.
I have thoroughly reviewed the manuscript. I believe the authors addressed all the issues raised by the reviewer satisfactorily. I have one more comment: line 353 - "4.1. T Pb, Cd, and..." - please delete "T"

Author Response

Point1. I have thoroughly reviewed the manuscript. I believe the authors addressed all the issues raised by the reviewer satisfactorily. I have one more comment: line 353 - "4.1. T Pb, Cd, and..." - please delete "T".
Response 1. Thanks for the observation. The letter T has been removed. It was a typo.

Reviewer 4 Report

The introduction section can also be appropriately deleted

Minor editing of English language required

Author Response

We appreciate the time the reviewers have taken to improve the quality of the manuscript. We have complied with lifting the comments, and we hope that the manuscript meets the readers expectations.